# Interfacial solute flux promotes emulsification at the water|oil interface

Guillermo S. Colón-Quintana[1,3], Thomas B. Clarke [1,3] & Jeffrey E. Dick [1,2]

Emulsions are critical across a broad spectrum of industries. Unfortunately, emulsification requires a significant driving force for droplet dispersion. Here, we demonstrate a mechanism of spontaneous droplet formation (emulsification), where the interfacial solute flux promotes droplet formation at the liquid-liquid interface when a phase transfer agent is present. We have termed this phenomenon fluxification. For example, when $HAuCl_4$ is dissolved in an aqueous phase and $[NBu_4][ClO_4]$ is dissolved in an oil phase, emulsion droplets (both water-in-oil and oil-in-water) can be observed at the interface for various oil phases (1,2-dichloroethane, dichloromethane, chloroform, and nitrobenzene). Emulsification occurs when $AuCl_4^-$ interacts with $NBu_4^+$, a well-known phase-transfer agent, and transfers into the oil phase while $ClO_4^-$ transfers into the aqueous phase to maintain electroneutrality. The phase transfer of $SCN^-$ and $Fe(CN)_6^{3-}$ also produce droplets. We propose a microscopic mechanism of droplet formation and discuss design principles by tuning experimental parameters.

Emulsions have long been shown to be useful in both industrial and pharmaceutical settings for drug development and manufacture[1-4]; however, much attention has more recently brought focus to the improvement of methods for emulsification. Numerous means of producing emulsions have been reviewed[5-8]. Energy can be supplied to mix two immiscible phases (e.g., water and oil) that are initially at equilibrium and form an emulsion consisting of water droplets in oil or oil droplets in water. However, unless a surface-active species is present to stabilize the droplets, droplets can rapidly coalesce as the system reverts to two bulk solutions[9]. Instead of supplying external energy, placing two immiscible phases together that are not in equilibrium can also induce emulsification due to the gradients in chemical potential[10]. Methods of this type have been termed "spontaneous emulsification" or "self-emulsification," and numerous reports have been published describing such phenomena[6,10]. Surfactants can be used to decrease the interfacial energy sufficiently such that emulsion droplets can spontaneously form[7]. Alternatively, amphiphilic solvents or solutes that are partially soluble in both phases can induce emulsification within ranges of specific molar ratios[11,12].

In this manuscript, we demonstrate that spontaneous emulsification at the liquid|liquid interface can be achieved using a well-

known phase transfer agent that induces significant anion flux between the oil and aqueous phases. One means of achieving such a high degree of flux is by having a relatively hydrophobic anion in the water phase, a relatively hydrophilic anion in the oil phase, and a phase-transfer agent, like tetraalkylammonium cation, in the oil phase. We have found spontaneous emulsification for a number of different transferring ions, for example, if $AuCl_4^-$ is present in the aqueous phase and tetrabutylammonium perchlorate ($[NBu_4][ClO_4]$) is present in the 1,2-dichloroethane (DCE) phase, the tetrabutylammonium cation can facilitate the transfer of anions across the two phases. Our results demonstrate that sufficient flux of ions across this interface can induce emulsification at the liquid|liquid boundary. This system is in contrast to the one used by Kaminska, et al., who showed that a microemulsion formed at the toluene|water interface when $HAuCl_4$ was present in the water and tetraoctylammonium bromide was present in the toluene phase[13]. The focus of that report was on the electrodeposition of gold nanoparticles with no focus on the mechanism of emulsification. In this study, we elucidate the mechanism of spontaneous emulsification by examining how the degree of the partitioning of $AuCl_4^-$ (or other aqueous ions) influences the extent of emulsification.

[1]Department of Chemistry, Purdue University, West Lafayette, IN 47907, USA. [2]Elmore Family School of Electrical and Computer Engineering, Purdue University, West Lafayette, IN 47907, USA. [3]These authors contributed equally: Guillermo S. Colón-Quintana, Thomas B. Clarke. ✉e-mail: jdick@purdue.edu

## Results

Chloroaurate has been shown to strongly partition from the aqueous phase to the DCE oil phase[14]. This can be observed by pipetting 4 mL of a DCE phase containing 0.1 M tetrabutylammonium perchlorate ([NBu$_4$][ClO$_4$]) into a glass vial and then pipetting an equal volume of an aqueous solution of 10 mM HAuCl$_4$ and 1 M KCl down the side of the same vial. Since DCE is denser than water, the DCE phase remains at the bottom of the vial, while the aqueous phase stays at the top. Within a minute of contact between the two phases, the aqueous phase, initially having a yellow color due to the chloroaurate ion, starts to lose its color and become cloudy, while the DCE phase begins to appear yellow. After about 120 min, the aqueous phase appeared entirely clear and the DCE phase became dark yellow (see Supplementary Fig. 1). We have previously quantified the partition coefficient of chloroaurate into the DCE phase under these conditions using inductively coupled plasma-mass spectrometry and found $K_P = 930$[14].

The bulk anionic transfer of chloroaurate from the water to the oil phase necessitates another ion crossing the phase boundary to maintain electroneutrality. Based on ion transfer potentials, the transfer of perchlorate ions from the DCE phase to the aqueous phase would be the most likely process (as opposed to proton or potassium ions transferring from the aqueous phase to the DCE). The Gibbs free energy of ion transfer and the ion transfer potentials for relevant ions in this manuscript (i.e., tetrabutylammonium, tetraethylammonium, hydronium, sodium, potassium, perchlorate, bromide, chloride, and hexafluorophosphate) are provided in Supplementary Table 1. The free energy values for aqueous-phase anions that phase transfer are difficult to measure and are complicated due to the ion pairing mechanism at the boundary; however, their partition coefficients, central to their interfacial flux, can be measured readily and are reported in the Supplementary Information File. This coupled ionic transfer must occur across the liquid|liquid boundary that is formed between these two liquid phases, but few studies have examined what happens at this interface during ion transfer in detail.

In this study, liquid|liquid interfaces were created by pipetting microliter-sized droplets of water and DCE between a glass slide and a cover slip (see Fig. 1). When the water phase contained HAuCl$_4$ and the DCE phase contained [NBu$_4$][ClO$_4$], an emulsion began to form at the liquid|liquid interface, as was observed with transmission light microscopy. If either HAuCl$_4$ or [NBu$_4$][ClO$_4$] is not present, no emulsion was observed.

As can be shown in Fig. 2, when DCE contains 0.1 M [NBu$_4$][ClO$_4$] and the aqueous phase contains 10 mM HAuCl$_4$ and 1 M NaCl, DCE droplets begin to form in the aqueous phase as early as 10 s after the two phases come into contact. These DCE droplets appear to grow over time, often via coalescence, where two DCE droplets fuse to form a larger droplet. Occasionally we observed DCE droplets in the aqueous phase coalesce with the bulk DCE phase. To characterize the extent of the emulsion at different points in time, droplets were identified within the same region of the image, and their areas and distances from the liquid|liquid boundary were measured using ImageJ (see Supplementary Fig. 2). This method was used to compare the extent of emulsification under different conditions in this study. Supplementary Fig. 3 shows the number of identified droplets as a function of distance from the interface at four different time points in the experiment. Over time, a greater number of droplets form and do so further away from the liquid|liquid boundary. As is seen in Supplementary Fig. 3, the number of droplets within 10 μm of the interface decreases with time due to coalescence. This coalescence can occur both between neighboring droplets and between emulsion droplets and the bulk DCE phase. This coalescence was visually observed and substantiated by the corresponding increase in the average cross-sectional area of droplets within this range (Supplementary Fig. 3). At all time points, the average droplet area decays with increasing distance from the interface.

To characterize the stability and reproducibility of the observed droplets, dynamic light scattering (DLS) measurements were performed on the droplets that were spontaneously formed. To achieve this, 10 mL of 10 mM HAuCl$_4$ was pipetted over 10 mL of DCE containing 0.1 M [NBu$_4$][ClO$_4$]. An overhead stirrer was inserted into the aqueous phase and was used to induce convection in the aqueous phase without making contact with the liquid|liquid boundary. This allowed the spontaneously formed DCE droplets to be suspended in the aqueous phase. DLS measurements were performed on this suspension for over an hour, and the droplet sizes are reported in Supplementary Fig. 4. These results show that the droplets are stable for over an hour, with monomodal distributions decreasing in diameter from just over 1 μm to <1 μm within the hour. Additionally,

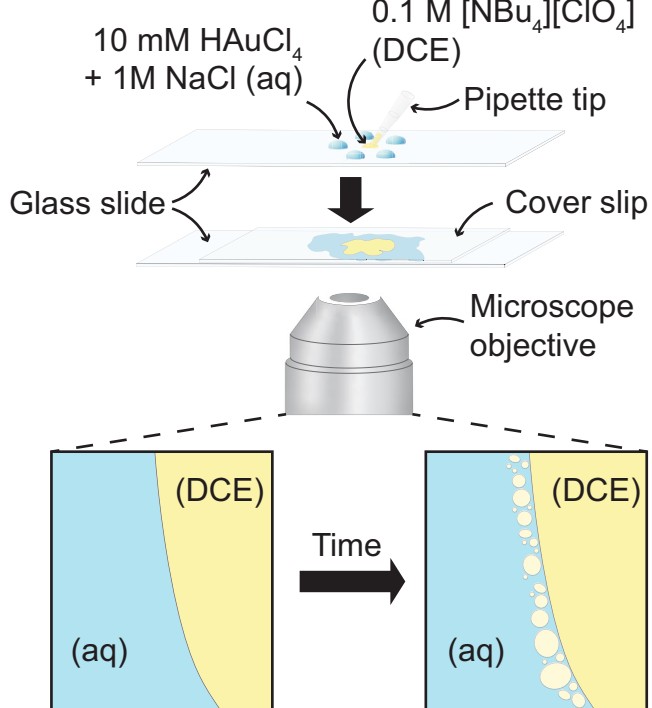

**Fig. 1 | Overview of the experimental setup.** Five 0.4 μL droplets containing 10 mM HAuCl$_4$ and 1 M NaCl are pipetted onto a glass slide with a DCE droplet containing 0.1 M [NBu$_4$][ClO$_4$] pipetted into the center. A cover slip is placed on top of the droplets and transmission light microscopy shows the spontaneous emulsification and generation of DCE droplets in the aqueous phase over time.

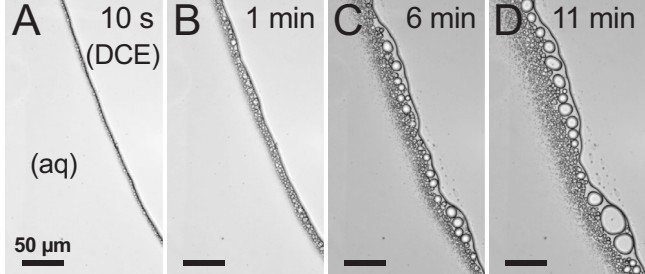

**Fig. 2 | Light microscopy images of the liquid|liquid interface over time.** Light microscopy images of the 10 mM HAuCl$_4$ + 1 M NaCl (aq)|0.1 M [NBu$_4$][ClO$_4$] (DCE) interface at **A** 10 s, **B** 1 min, **C** 6 min, and **D** 11 min after initial contact of the two phases. Scale bar for all images is 50 μm. All optical micrographs were taken with a ×40 NA 0.60 objective and a 500 ms exposure time.

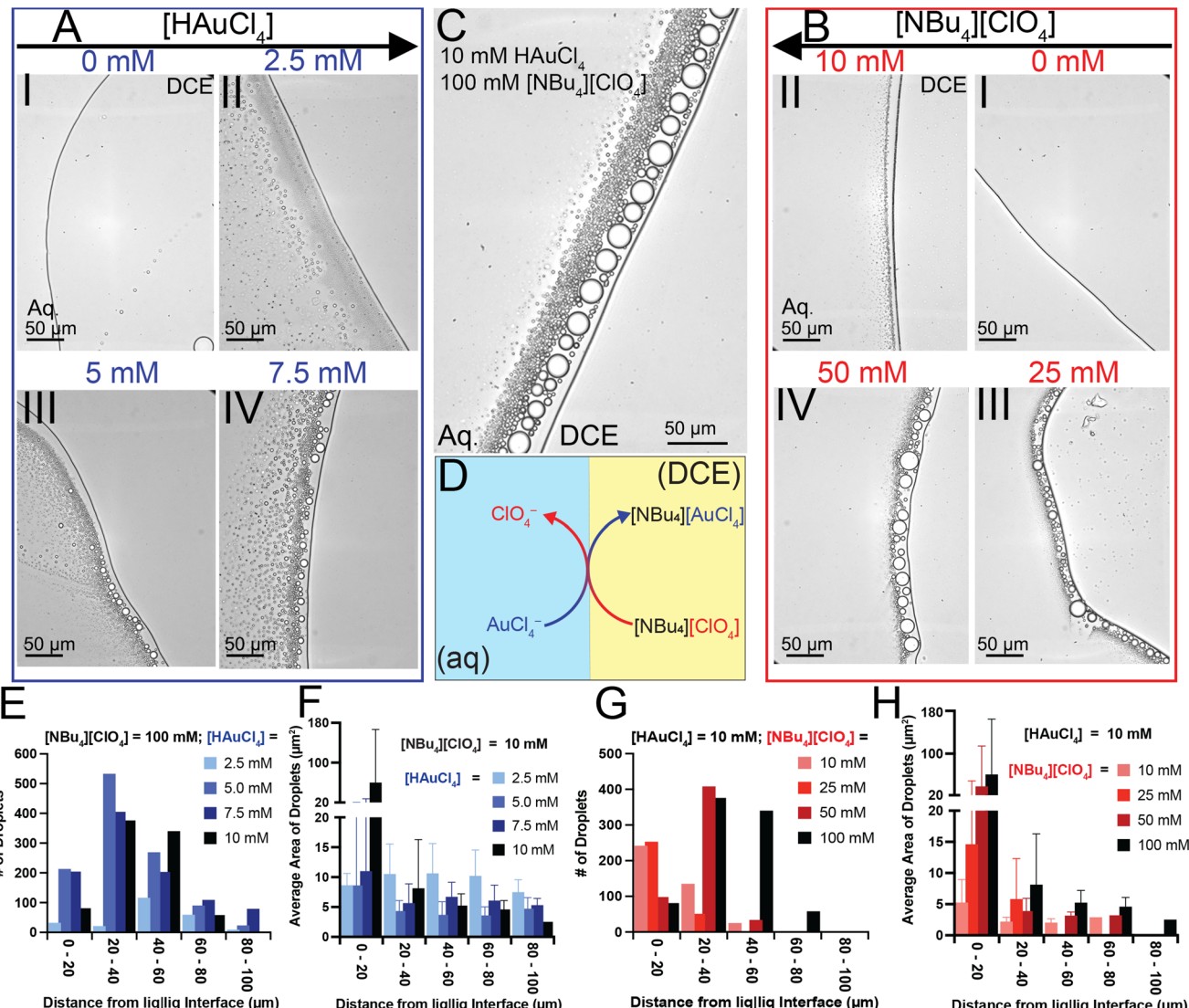

**Fig. 3 | Extent of emulsification as a function of transferring ion concentrations. A** Optical micrograph of the effect of HAuCl₄ concentration on droplet formation. An increase in concentration can be seen sequentially for I (0 mM), II (2.5 mM), III (5 mM), and IV (7.5 mM). The concentration of [NBu₄][ClO₄] (DCE) was kept constant at 100 mM for these images. **B** Optical micrographs of the effect of [NBu₄][ClO₄] concentration on droplet formation. An increase in concentration can be seen sequentially for I (0 mM), II (10 mM), III (25 mM), and IV (50 mM). The concentration of HAuCl₄ (aq) was kept constant at 10 mM for these images. **C** Optical micrograph for the droplet formation at 10 mM HAuCl₄ and 100 mM [NBu₄][ClO₄]. **D** Schematic representation of the proposed mechanism for partitioning of chloroaurate from aqueous to organic media in the presence of

NBu₄⁺ while maintaining electroneutrality with ClO₄⁻. Histograms presented below schematic show the effect of changes in concentration on the frequency (**E, G**) and average cross-sectional area (**F, H**) of droplets as a function of distance from the interface for chloroauric acid (Blue; **E** and **F**) and tetrabutylammonium perchlorate (Red; **G** and **H**), respectfully. The error bars in **F** and **H** correspond to standard deviations about the mean for N equal to the number of droplets identified in that region (from **E** and **G**). All images and data for histograms was gathered 10 min after initial contact of aqueous and DCE phases. All optical micrographs were taken with a ×40 NA 0.60 objective and a 500 ms exposure time. Source data are provided as a Source Data file.

images were taken of the two bulk phases during flux-induced emulsification without added convection, and the aqueous phase appeared cloudy due to the emulsion for a few hours after initial contact (see Supplementary Fig. 5). These results suggest that these droplets can be stable for long periods of time. Furthermore, DLS measurements of four emulsions prepared under the same conditions (10 mM HAuCl₄ (aq) and 0.1 M [NBu₄][ClO₄] (DCE)) and taken at the same time after initial solution contact (5 min) gave reproducible droplet sizes with the average droplet diameter = 1390 nm and an RSD of 12.3% (Supplementary Fig. 6 and Supplementary Table 2).

To elucidate a mechanism for this observed emulsification, we removed or changed the nature or concentration of various

constituents. First, to ensure that the spontaneous emulsification was not simply due to the dissolution of water into DCE and vice versa, mutually saturated solutions were tested (Supplementary Fig. 7). When a water-saturated DCE phase containing 0.1 M [NBu₄][ClO₄] was used as the organic phase and a DCE-saturated aqueous solution containing only 1 M NaCl was used for the aqueous phase, no emulsification was observed. It was only when HAuCl₄ was also present in the DCE-saturated aqueous solution that emulsification spontaneously occurred (Supplementary Fig. 7). The latter results showed no noticeable differences in emulsion behavior when compared to non-saturated solutions, suggesting that the degree of saturation of our solutions prior to contact does not play a critical role in the formation of the emulsion. Secondly, changing the added aqueous electrolyte from 1 M

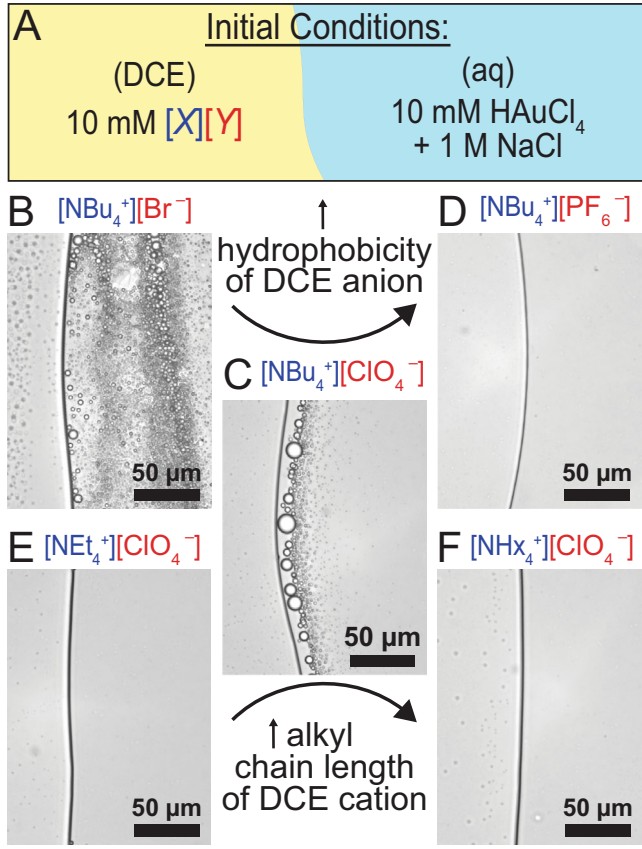

**Fig. 4 | Effect of organic-phase electrolyte on emulsification behavior. A** Graphic showing initial conditions where the DCE phase contains 10 mM of a non-aqueous salt with variable cation ($X$) and anion ($Y$), while the aqueous phase contains 10 mM HAuCl$_4$ + 1 M NaCl. **B–F** Light microscope images showing the liquid|liquid interface after 10 min from initial contact of the two phases. All images show the DCE phase on the left and the aqueous phase on the right. For **B–D**, tetrabutylammonium (NBu$_4^+$) salts were used in the DCE with the anion being Br$^-$ (**B**), ClO$_4^-$ (**C**), or PF$_6^-$ (**D**). For **E** and **F**, perchlorate salts were used in the DCE with the cation being tetraethylammonium (NEt$_4^+$, **E**) or tetrahexylammonium (NHx$_4^+$, **F**). All optical micrographs were taken with a ×40 NA 0.60 objective and a 500 ms exposure time.

NaCl to 1 M KCl shows identical results (see Supplementary Fig. 8), indicating that changing the supporting electrolyte has a minimal effect. However, removing the NaCl or KCl entirely from the aqueous phase shows emulsification on both sides of the liquid|liquid interface, with water droplets being observed in DCE, in addition to DCE droplets formed in water (see Supplementary Fig. 9). This will be discussed in more detail below.

When either the HAuCl$_4$ or the [NBu$_4$][ClO$_4$] were removed from the aqueous or DCE phases, respectively, no detectable emulsification was observed (see Fig. 3A(I), B(I)). As the concentrations of these species are individually increased, a proportional increase in emulsification was observed (see Fig. 3). Either the HAuCl$_4$ concentration was varied from 0 to 10 mM, with the [NBu$_4$][ClO$_4$] concentration being held constant at 100 mM (Fig. 3A) or the HAuCl$_4$ concentration was held constant at 10 mM and the [NBu$_4$][ClO$_4$] concentration was varied from 0 to 100 mM (Fig. 3B). When the concentrations of HAuCl$_4$ was 10 mM and [NBu$_4$][ClO$_4$] was 100 mM, the greatest amount of emulsification was observed (Fig. 3). All images were taken 10 min after solutions came into contact to standardize and compare emulsions within similar timeframes. A clear dependence on the concentration of both AuCl$_4^-$ and [NBu$_4$][ClO$_4$] was observed quantitatively as shown by Fig. 2E, G, where greater degrees of emulsification were observed at higher concentrations. Examining the number of droplets identified at

different concentrations does not provide a clear summary of the emulsification (for example, a greater total number of droplets were identified for the case where [HAuCl$_4$] = 5 mM as compared to other concentrations of HAuCl$_4$). However, examining the cross-sectional areas sheds light on this observation. The average cross-sectional area of droplets was the largest under the conditions of the highest concentrations of HAuCl$_4$ or the [NBu$_4$][ClO$_4$] (Fig. 3F, H). So despite forming a greater number of droplets when [HAuCl$_4$] = 5 mM, a majority of these droplets are smaller than the droplets at other concentrations, as indicated by the average cross-sectional areas. Due to the fact that droplets coalesce, analyzing the droplet counts alone is insufficient to characterize the extent of emulsification. For all cases, a size gradient was observed as a function of distance from the interface, wherein larger droplets formed closer to the liquid|liquid interface and smaller droplets extended into the bulk aqueous phase. The concentration dependence signifies that flux plays a critical role in emulsification.

Results presented in Fig. 3 suggest that the degree of emulsification is directly related to the concentrations of HAuCl$_4$ and [NBu$_4$][ClO$_4$]. These observations, in addition to the fact that AuCl$_4^-$ partitions into the DCE phase under these conditions imply that the degree of AuCl$_4^-$ partitioning controls the degree to which an emulsion forms. As previously mentioned, if AuCl$_4^-$ partitions, it must be accompanied by an anion from the DCE phase transferring to the aqueous phase. In this case, perchlorate transfers into the aqueous phase to maintain electroneutrality. Therefore, we propose that the mechanism for emulsification is due to the reaction is shown in Fig. 3D and is represented by the forward reaction of the following equilibrium:

$$AuCl_4^-(aq) + ClO_4^-(DCE) \rightleftharpoons AuCl_4^-(DCE) + ClO_4^-(aq) \qquad (1)$$

The role of NBu$_4^+$ as a phase-transfer agent cannot be overlooked. Phase transfer agents, such as quaternary ammonium salts and phosphonium salts, have shown an innate ability to transfer reactants between immiscible phases to promote reactions that would normally be difficult[15]. These phase transfer agents can facilitate reactions by promoting changes in solubility and activating a reactant prior to the reaction. Tetrabutylammonium, for example, has been previously shown to facilitate the transfer of bromide across an interface to react with benzene chloride to produce benzene bromide[16]. For the work presented here, tetrabutylammonium perchlorate can be shown to have a similar phase transfer effect. Figure 3D reflects how NBu$_4^+$ facilitates the partitioning of AuCl$_4^-$.

We were also interested in probing the degree of emulsification based on the salt present in the oil phase. Figure 4A is a schematic that shows the various ions that can be changed. Changing the anion that transfers into the aqueous phase (Fig. 4B, D) to maintain electroneutrality had a dramatic effect compared to the control (Fig. 4C): the more hydrophilic the anion, the more rapidly the droplets formed. A more hydrophobic anion, PF$_6^-$, has the least favorable free energy change upon transferring from DCE to water (Supplementary Table 1), which disfavors the flux of ions and prevented emulsion formation[17]. Changing the alkylammonium salt had a less dramatic effect: tetraethylammonium perchlorate disallowed droplet formation likely because the salt is quite soluble in water. Droplets did not form in the presence of tetrahexylammonium perchlorate likely because of the hydrophobicity of the alkylammonium salt. Large carbon chains may be drawn to organic solvents but do not provide the necessary activation for reaction[15]. For larger carbon chains, Aoki and coworkers have shown that as the length of the quaternary ammonium chains increases the emulsion droplet stability is decreased and emulsification is lessened[18]. Our results seem to agree with these observations, as can be seen in

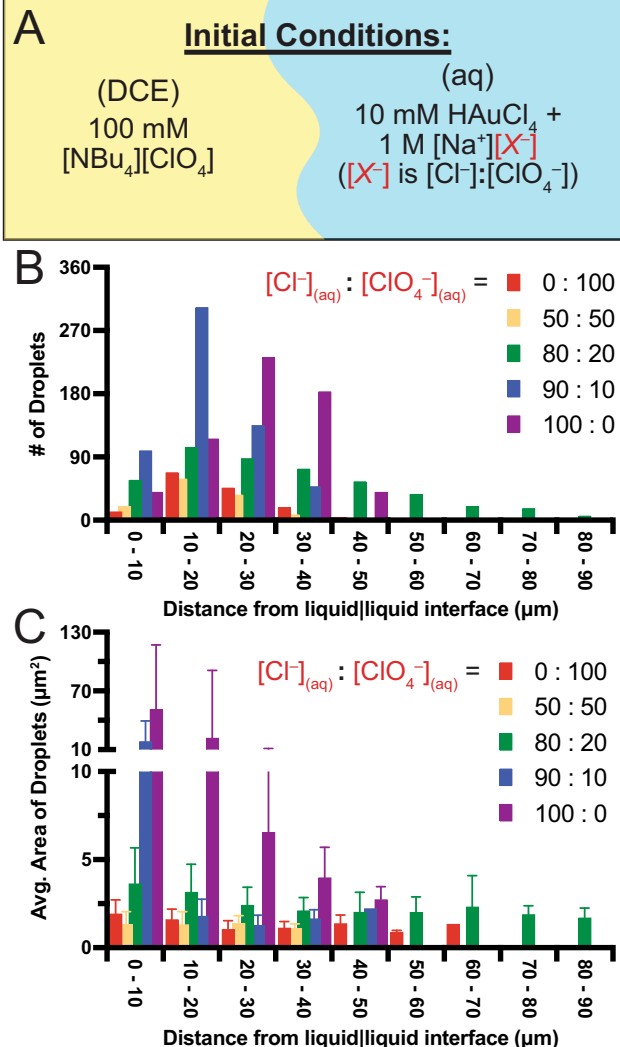

**Fig. 5 | Degree of disequilibrium affects the extent of emulsification. A** Graphic showing initial conditions where the DCE phase contains 100 mM of a tetrabutylammonium perchlorate, while the aqueous phase contains 10 mM HAuCl₄ with different ratios of sodium chloride and sodium perchlorate (but always 1 M total supporting electrolyte concentration). Histograms show the number of emulsion droplets measured (**B**) and the average area of measured droplets (**C**) as a function of distance from the liquid|liquid interface. All droplets were measured 10 min from the initial time of contact. Ratios of $[Cl^-]_{(aq)}$:$[ClO_4^-]_{(aq)}$ used were 0:100 (red), 50:50 (yellow), 80:20 (green), 90:10 (blue), and 100:0 (purple). The error bars in **C** are standard deviations about the mean for N equal to the number of droplets identified in that region (from **B**). Source data are provided as a Source Data file.

Fig. 4, where the emulsification can be seen for $NBu_4^+$ but not for other quaternary ammonium salts.

We further tested our flux hypothesis by adding $ClO_4^-$ into the aqueous phase to influence the equilibrium and disfavor the anionic transfer reaction (Fig. 5A). To maintain similar electrolyte concentrations, we maintained the aqueous supporting electrolyte concentration at 1 M but used various ratios of sodium chloride and sodium perchlorate (Fig. 5A). As can be seen in the histograms presented in Fig. 5B, C, the number and average cross-sectional area of DCE emulsion droplets measured in the aqueous phase decreases as the ratio of $[Cl^-]_{(aq)}$:$[ClO_4^-]_{(aq)}$ is decreased. The microscope images after 10 min of solution contact are shown in Supplementary Fig. 10. In other words, if a greater amount of $ClO_4^-$ is already present in the aqueous phase, the initial conditions of the experiment will be closer to the equilibrium of

Eq. (1), and the reaction will proceed to a lesser degree. This decrease in reactivity, therefore, causes less flux at the interface and decreases the extent of emulsification. Some groups have claimed that spontaneous emulsification can be achieved when a common ion is present in both the aqueous and oil phases[18,19]. In contrast, we see a decrease in emulsification when perchlorate is present in both phases because the presence of perchlorate in the aqueous phase disfavors the anionic transfer reaction. These results are consistent with Le Chatelier's principle and further indicate that flux is the main driver of emulsification.

## Discussion

In this section, we use lessons learned from the chloroauric acid system described above to develop a more generalized platform for emulsification. Partitioning ions between the water and oil phase follow an equilibrium reaction like that of Eq. (1), but other anions can be used instead of $AuCl_4^-$. In addition to using HAuCl₄ in the aqueous phase, we also used KSCN and $K_3[Fe(CN)_6]$ as a source of different partitioning anions to induce emulsification (Fig. 6). Tetraalkylammonium cations have been shown to transfer thiocyanate ions from water to oil[15], and ferricyanide has similarly been shown to transfer to the DCE phase in the presence of $[NBu_4][ClO_4]$[20].

From Supplementary Fig. 9, we have found that the presence of an additional aqueous electrolyte (NaCl or KCl) does not seem to influence the DCE droplet formation in the aqueous phase but does influence whether water droplets can form in the DCE phase. This can be observed as well in Fig. 6A. To explain this case, it is important to consider the solvation of the transferring aqueous anion and that transferring ions bring water molecules with them. For example, molecular dynamics simulations have shown that ions transferring across liquid|liquid boundaries bring solvent molecules with them[21–23]. When 1 M KCl is present in the water, the $AuCl_4^-$ will become less solvated in the aqueous phase and will transfer with fewer solvent molecules because of solvation competition with other ions in greater excess. When KCl or NaCl is absent in the aqueous phase, more water molecules can transfer with the anions such that water droplets can form in the DCE. This is also the case when other aqueous anions ($SCN^-$ and $[Fe(CN)_6]^{3-}$) transfer into the DCE phase. Interestingly, DCE droplets do not form in the aqueous phase when using KSCN or $K_3[Fe(CN)_6]$ (Fig. 6B, C). This is likely because DCE droplets can be stabilized in some cases by an antagonistic salt (as is the case when using HAuCl₄)[24]. This is represented in our proposed microscopic mechanism shown in Fig. 6D.

Furthermore, to show generalizability, we used HAuCl₄ and $[NBu_4][ClO_4]$ to induce emulsification for various water|oil interfaces. While all observations thus far have been with water and DCE, we also dissolved 0.1 M $[NBu_4][ClO_4]$ into dichloromethane, chloroform, and nitrobenzene. Images of the interface of these solutions in contact with aqueous solutions of 10 mM HAuCl₄ show similar behavior to the results with DCE for cases both with and without added NaCl in the aqueous phase (Supplementary Fig. 11).

All of our observations to this point allow us to propose a generalized microscopic model for the flux-induced emulsification, as shown in Fig. 6D. While the emulsification is driven by the partitioning of ions at the boundary and flux of ions to the boundary, the mechanism of ion transfer is more involved. Our results suggest that when ions transfer across the interface, ion pairs can form that can stabilize curved interfaces. Such ion pairs are called antagonistic salts[24]. In the case of chloroauric acid in the water, tetrabutylammonium chloroaurate forms at the liquid|liquid interface. In a complex matrix, molecules that have the highest affinity for the boundary will adsorb to the boundary. Thus, we expect droplets formed in either phase to be stabilized by the same antagonistic salt. The sinuosity of the liquid|liquid interface has been shown, and

transferring ions increases the surface roughness of this interface, namely via "water fingers"[25]. Ions that transfer from water to oil have been shown to bring water molecules with them, forming a finger-like structure; whereas the transfer of oil-phase ions into water has not been shown to bring in solvent molecules and instead is facilitated by water fingers that can engulf oil-phase ions[21]. Thus, substantial ion transfer can create morphologies like that shown in the middle panels in Fig. 6D due to many water fingers. Protrusions of water fingers can pinch off upon further ionic flux in all cases. While the size of the newly formed droplets is difficult to know at the time of their initial formation, these droplets have been observed to coalesce, thereby forming larger droplets over time. Thus, the initial droplet sizes relative to the water fingers in Fig. 6D are not necessarily drawn to scale. However, in cases where there is a sufficiently high degree of ion flux, the density of water fingers is high enough such that water can enclose regions of oil to form oil droplets. This is likely why $AuCl_4^-$, which strongly partitions into the DCE phase, creates DCE droplets in water (Fig. 6A), whereas other ions with lower partition coefficients, like $SCN^-$ and $Fe(CN)_6^{3-}$, cannot form dense enough regions of water fingers to enclose oil droplets and instead only produce water droplets (Fig. 5B, C). We have measured the partition coefficient of $SCN^-$ from water to DCE to be 0.07 via a colorimetric experiment (Supplementary Fig. 12) and the partition coefficient of $Fe(CN)_6^{3-}$ to be 0.03 via a voltammetric experiment (Supplementary Fig. 13). These partition coefficients are orders of magnitude smaller than the corresponding partition coefficient for $AuCl_4^-$, which explains the lack of DCE droplets formed in these cases. Our model is reminiscent of a meandering river that creates oxbow lakes over time[26].

While our model predicts droplets form in both phases, the fate of those droplets depends strongly on how well the antagonistic salt stabilizes a droplet in a given phase. For our results with KSCN and $K_3[Fe(CN)_6]$, no DCE droplets are observed in the aqueous phase. Our model predicts that tetrabutylammonium thiocyanate is not an effective stabilizer of DCE droplets in water. Furthermore, $Fe(CN)_6^{3-}$ is the most hydrophilic example where one can observe flux-induced emulsification. Thus, another important aspect of our model is the solvation of the transferring ions. Ferricyanide will be much more hydrated, which can increase the probability of water droplet formation.

Given our model and observations, we offer the following design principles for flux-induced emulsification (fluxification):

1. Only systems that are amenable to a phase transfer agent will allow for emulsification.

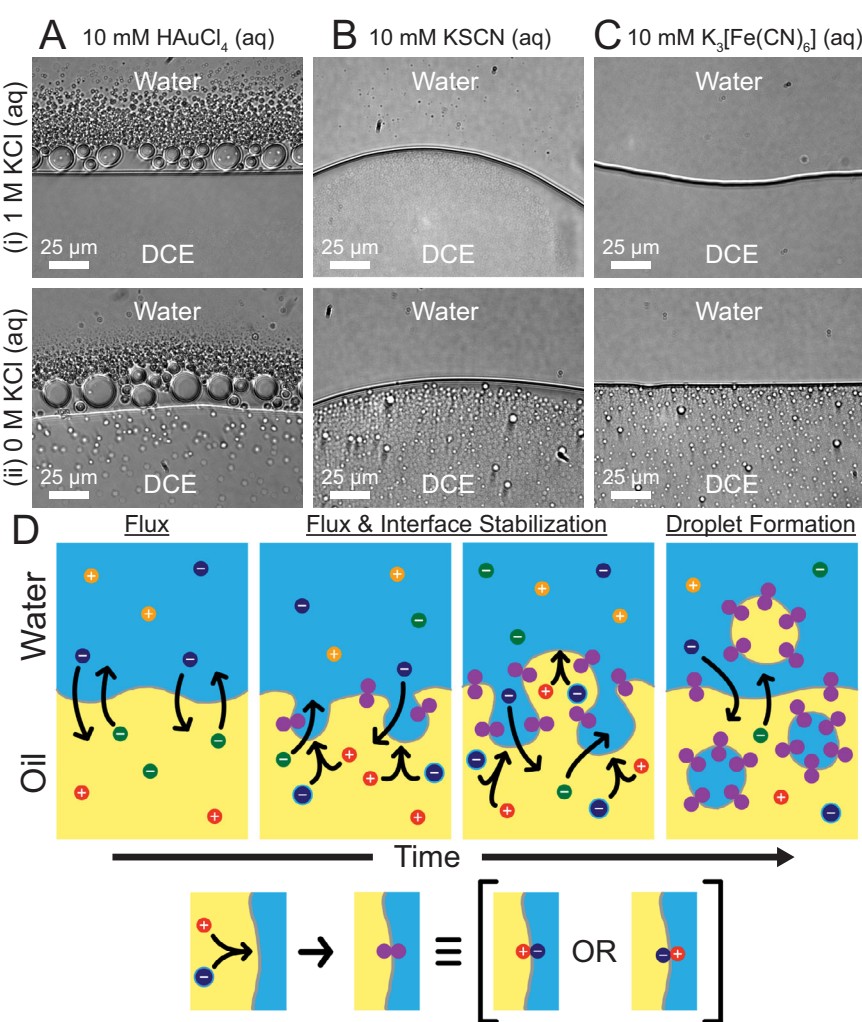

**Fig. 6 | Emulsification achieved with different transferring ions, the effect of aqueous salt concentration, and proposed microscopic model of flux-induced emulsification.** Light microscopy images were taken at the water|DCE boundary when the aqueous phase contained 10 mM HAuCl₄ (**A**), KSCN (**B**), or K₃[Fe(CN)₆] (**C**) with either 1 M KCl (**i**) or 0 M KCl (**ii**). The DCE phase always contained 0.1 M [NBu₄][ClO₄]. The water and DCE phases are indicated, and the scale bars are 25 µm. **D** Proposed microscopic model of how the transferring ions can cause water-in-oil droplets and/or oil-in-water droplets to form and be stabilized by antagonistic salts. All optical micrographs were taken with a ×40 NA 0.60 objective and a 500 ms exposure time.

2. The extent of droplet formation depends on the flux at the liquid|liquid interface: the higher the flux, the more quickly droplets will form. From Fig. 4, faster flux produces smaller droplets likely because droplets have not had enough time to coalesce.

3. The probability of droplet formation depends on the following parameters:
   a. Hydrophilicity/hydrophobicity and hardness/softness of the transferring ions.
   b. Solvation competition with other ions in solution.
   c. Ability of complex ions (antagonistic salts) to stabilize droplets in one phase over another. The orientation of the antagonistic salts at the boundary must also play a role (not considered in this work).

Given the rising interest in studying and engineering chemistry in microenvironments[27–29], our system offers a valuable alternative way to emulsify a water and oil phase without the use of large amounts of amphiphilic solvents or additional organic solutes. This system uses a phase-transfer agent to drastically increase the flux of an anion (e.g., $AuCl_4^-$) and the degree of emulsification can be tuned by the hydrophilicity and/or concentration of a secondary anion in the DCE solution. Furthermore, we can get water droplets in the oil phase or oil droplets in the aqueous phase by adding or removing additional electrolytes in the water phase. The ability to tune the extent of emulsification with the choice of electrolyte provides a robust alternative to spontaneous emulsification methods via more complex strategies. Finally, our results offer insight into a new physical means by which nature can spontaneously produce microcompartments, a topic of intense interest in understanding the abiotic origins of life[27,30–32].

# Methods
## Materials
All solutions were prepared with fresh solvents, aqueous solutions were prepared with ultra-pure water (18.20 MΩ cm) obtained from a Millipore GenPure water filtering system; organic solutions were prepared with 1,2-dichloroethane (DCE), dichloromethane, chloroform, or nitrobenzene obtained from Sigma-Aldrich. Organic salts such as tetrabutylammonium perchlorate (99% purity), tetrabutylammonium hexafluorophosphate (99% purity), tetrabutylammonium bromide (97% purity), tetrabutylammonium chloride (99% purity), tetraethylammonium perchlorate (97% purity) and tetrahexylammonium perchlorate (97% purity) were obtained through Sigma-Aldrich. Potassium thiocyanate and potassium hexacyanoferrate (III) were obtained from Thermo Scientific, whereas iron (III) nitrate was obtained from Aldrich. Sodium chloride (99% purity) from Fisher Bioreagents and sodium perchlorate (98% purity) obtained through Sigma-Aldrich were used for aqueous solutions. Solutions containing gold were based on chloroauric acid (99.995% purity) obtained through Sigma-Aldrich. All reagents used for the process here shown were of analytical grade and were used without further purification.

New glass coverslips were used for each experiment prior to each test and were obtained from VWR (Radnor, PA). A 24 × 40 mm cover glass was used on the bottom and a 25 × 25 mm cover glass was used on top. A gold ultramicroelectrode ($r = 6.25\,\mu$m) and Ag|AgCl reference electrode were obtained from CH Instruments (Austin, TX).

## Methods
### Microscopy experiments.
All microscopy experiments were performed on glass slides, onto which a pentagon of droplets of an aqueous solution containing $HAuCl_4$, KSCN, or $K_3[Fe(CN)_6]$ with or without NaCl or KCl were pipetted onto the surface of the glass slide.

An organic droplet (DCE, dichloromethane, chloroform, nitrobenzene) containing $[NBu_4][ClO_4]$ or other tetraalkylammonium salt was then pipetted onto the center of the pentagon of aqueous droplets. This allowed for the aqueous phase to surround the organic phase and maximize the contact area while minimizing the effects caused by evaporation. A cover slide was then placed on top of all droplets, making sure that solutions did not spread outside of the covered area. Finally, optical images were taken in transmission mode at timed intervals to allow for qualitative and quantitative assessment of microdroplet formation. This process was repeated for variations of concentrations of $HAuCl_4$, for different sources of transferring ions (KSCN or $K_3[Fe(CN)_6]$), for different ionic strengths of aqueous solution (with or without NaCl or KCl), for different organic salts, and for different oil phases.

All optical micrographs presented in the main text were taken with a Leica DMi8 inverted microscope obtained from Leica Microsystems (Germany). A TL LED Lamp 12 V DC max light source was used to illuminate glass slides in a transmission microscopy mode and was also obtained from Leica Microsystems. Images were taken with the use of a C15440 OrcaFusionBT sCMOS camera obtained from Hamamatsu Photonics (Japan). A ×40 objective with an NA of 0.60 was obtained from Leica Microsystems and was used for all micrographs shown. Images and optical videos were recorded in bright-field illumination with an exposure time of 500 ms.

### Droplet size measurements.
Dynamic light scattering (DLS) measurements of droplet sizes were performed using a Zetasizer Ultra instrument (Malvern Panalytical, Worcestire, UK). In these experiments, an aqueous $HAuCl_4$ solution was pipetted into an equal volume DCE solution containing 0.1 M $[NBu_4][ClO_4]$ in a scintillation vial. An overhead stirrer (Orion, Versa Star Pro benchtop meter, Thermo Scientific) was used for some experiments, as noted in the Supporting Information file, to induce convection in the aqueous phase for 1 min. Droplets were allowed to form for 5 or 10 min and then portions of the aqueous solution containing DCE droplets were pipetted from the scintillation vial to a 1 cm glass cuvette, which was inserted into the DLS instrument. The DLS measurement was taken for 60 runs at 1.64 s/run with 120 s equilibration time at an equilibration temperature of 25 °C. The polydispersity index was <0.1 for all measurements.

### Partition coefficient studies.
The partition coefficient of $SCN^-$ was measured experimentally via UV–VIS and the partition coefficient of $[Fe(CN)_6]^{3-}$ by use of electrochemical methods for ferricyanide. In both experiments, an equal volume of either 10 mM potassium thiocyanate or 10 mM potassium ferricyanide was added to a scintillation vial containing an equal volume of a DCE solution containing 0.1 M $[NBu_4][ClO_4]$ solution. These two phases were then vigorously mixed for 5 min to maximize partitioning between the two phases. Each vial was then set aside for about 2 h to allow the solutions to reach equilibrium and allow phase separation due to differences in density. For the thiocyanate experiment, a V-650 UV–VIS spectrophotometer (Jasco Inc., Japan) was used. The aqueous phase was removed from the vial and an excess of $Fe(NO_3)_3$ was added to form a red complex with the remaining thiocyanate that did not partition. The same was done for the 10 mM potassium thiocyanate solution used before contact with the organic solution. Absorbance measurements were made on these complexation solutions before and after partitioning. Additionally, absorbance measurements were made on stock solutions of known thiocyanate solutions with excess $Fe(NO_3)_3$ to create a calibration plot. The absorbance value of the potassium thiocyanate solutions before and after partitioning was then used to find the initial and final concentrations of thiocyanate. The ratio of these concentrations was then used to calculate the partition coefficient. This was repeated for an $N = 4$.

For the partitioning studies of $[Fe(CN)_6]^{3-}$, a CHI model 6284 potentiostat (CH Instruments, Austin, TX) was used to measure the concentration of ferricyanide before and after partitioning with 0.1 M $[NBu_4][ClO_4]$ DCE solution. A gold ultramicroelectrode ($r = 6.25\,\mu m$) was used as the working electrode, and an Ag|AgCl (1 M KCl) reference electrode, and a glassy carbon rod counter electrode were also used. Cyclic voltammetry was taken from $-0.1$ to 0.7 V vs. Ag|AgCl at a 50 mV/s scan rate before and after mixing. The concentrations of ferricyanide were obtained from the steady-state current due to ferricyanide reduction and the ratio of concentrations was taken to be the partition coefficient.

## Data availability
The data that support the findings of this study are available within the paper, the Supplementary Information file, and the Source Data file. Source data are provided with this paper.

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

## Acknowledgements
G.S.C., T.B.C., and J.E.D would like to acknowledge support from the National Institutes of Health under Grant No. R35-GM138133-01. We would like to thank Christophe Renault for his helpful discussions, as well as Rebecca Clark for her help in statistical analysis.

## Author contributions
G.S.C., T.B.C., and J.E.D devised the research plan and conceived experiments. G.S.C. and T.B.C. conducted the experiments. G.S.C., T.B.C., and J.E.D. wrote the manuscript. J.E.D. supervised all aspects of the work.

## Competing interests
The authors have filed a provisional patent application on the work described in this manuscript.
