## [Peer Review File · Nature Communications]

REVIEWER COMMENTS

Reviewer #1 (Remarks to the Author):

In this manuscript, the authors report spontaneous emulsification at the water/oil (1,2-dichloroethane, DCE) interface caused by the anion flux between the two phases. The interfacial flux of anions is enhanced by a phase-transfer agent (NBu₄⁺) present in the oil phase. The emulsification can be tuned by tuning the concentration and type of anions in the two phases. The emulsification mechanism was well discussed. The research topic is interesting and the results are reasonable. I recommend this manuscript to be published after minor revision.

Specific comments:

1. Page 3, for the partitioning test of chloroauric acid between the water and oil phases (Figure S1), KCl was used as the supporting electrolyte in the aqueous phase. Do KCl and NaCl have different effects on emulsification?
2. Figure S5, it seems that many droplets exist in the DCE phase. Can water droplets form in the oil phase when no NaCl is present in the water phase?
3. Page 10, line 200, "ClO₄⁻ (B), Br⁻ (C)" is incorrect?
4. Page 12, line 222, it is better to write "the number of emulsion droplets" instead of "the number of microemulsion droplets".

Reviewer #2 (Remarks to the Author):

This manuscript by Colón-Quintana et al. demonstrates a method of emulsification at the oil/water interface using solute flux enhanced by a phase transfer reagent. They achieve emulsification using tetrachloroaurate as the solute in the aqueous phase and tetrabutylammonium cation as the phase transfer agent in the oil phase. When the aqueous phase containing HAuCl₄ comes into contact with an oil phase containing [NBu₄][ClO₄], [NBu₄]⁺ ions facilitate the transfer of AuCl₄⁻ ions into the oil phase. To maintain electroneutrality in the aqueous phase, ClO₄⁻ anions are transferred into the aqueous phase. The degree of emulsification was found to be directly correlated to ion concentrations. The emulsification process was also sensitive to the hydrophobicity of cations and anions in the oil phase.

Utilizing solute flux as the driving force for emulsification is certainly interesting. Emulsification is characterized here using optical microscopy, and I found the optical microscopy images quite clear and easy to follow. The experiments that show the effect of the identity of organic phase ions are also nice and convincing. However, in its current form, the manuscript does not provide a clear understanding of flux-induced emulsification. Moreover, the authors have not provided enough background for the reader to understand their choice of reagents and their mechanistic role in the emulsification process. If both of these concerns are addressed, I believe the results in the manuscript can be significant for a wider audience. In the following, I list my specific questions and suggestions:

1. The criteria for the choice of AuCl₄⁻ as the aqueous phase anion needs to be stated. If flux-induced emulsification is a general phenomenon, what are the relevant parameters to consider when choosing the aqueous phase anion? If the aqueous phase contained other large, weakly hydrated anions such as SCN⁻, would they also give rise to emulsification?
2. Similarly, how are the required concentrations of the aqueous phase and organic phase ions and the aqueous electrolyte (1 M) decided? To elaborate this question further, The concentration series in Figure 2 uses 0, 2.5, 5, and 7.5 mM of HAuCl₄ and 0, 10, 25, and 50 mM [NBu₄][ClO₄] – why not use a 1:1 ratio of cations and anions? How were the concentrations selected? Could authors also comment on the role of 1M NaCl in this process?
3. It needs to be made clear how the droplets are formed and stabilized here. For example, if we compare this method to a more well-known emulsification process, such as surfactant-based emulsification, how are these droplet interfaces different? Do the cations and anions both occupy the surface of the droplets in a double-layer fashion? The droplets are always formed on the

aqueous side: How are the droplets transferred to that side?

4. How long are the droplets stable? Some information on the time stability of these emulsions needs to be added.

5. Figure 2 shows the effect of the anions and cations' concentration dependence on emulsification efficiency. To quantify emulsification efficiency, authors have used histograms of the number of droplets versus distance from the liquid/liquid interface. However, these plots do not show a proportional emulsification increase with increasing ionic concentration. For example, according to Figure 2E, the total number of droplets is higher at 5 mM H₂AuCl₄ than 10 mM H₂AuCl₄, which contradicts the authors' conclusion. Moreover, it is not clear if this type of histogram is the appropriate way of analysis here. A plot of the total number of droplets as a function of concentration would be more informative.

6. Figure 2F & H: what do the error bars represent?

7. Some details on the emulsification process would be useful for reproducing the results: How reproducible are the droplet counts as a function of concentration? (Can the authors provide error bars on these histograms?) Does the order of mixing of the aqueous and organic phases influence the result?

8. On page 4, line 80, the authors state, "Based on ion transfer potentials, the transfer of perchlorate ion from the DCE phase to the aqueous phase would be the most likely process (as opposed to proton or potassium ions transferring from the aqueous phase to the DCE). Can the authors provide the ion transfer potential values of all the cations and anions involved (including the supporting electrolyte) to support this argument? A table with all these values will make it much easier for the reader to understand the claim.

9. On page 6, line 111, "At all-time points, the average droplet area decays with decreasing distance from the interface." This should be: increasing distance

10. Schematic 1 shows 1M NaCl in the aqueous phase, whereas the text (page 3, line 71) says it is 1M KCl – this needs to be corrected.

11. The experiments by varying the hydrophobicity of the organic phase anion and cations are nicely done. Can the authors provide general guidelines for choosing the appropriate ionic species (both in the aqueous and organic phase) for a given water/oil interface (perhaps based on oil/water partition coefficients or solvation free energies?). Adding such a section would make the manuscript relevant for a wider audience.

Reviewer #3 (Remarks to the Author):

Colón-Quintana et al. report in their study on spontaneous formation of emulsion using AuCl₄⁻ as a phase transfer agent. The authors provide a parameter study where they varied the concentration of the "emulsifier" and performed some control experiments. Emulsification was measured by recording light microscopy images close to the oil-water interface.

I believe this study provides information on an intriguing phenomenon that is highly interesting from a fundamental aspect. The authors' experiments helped to show that AuCl₄⁻ is important for the self-emulsification. However, they could not reveal the molecular mechanism behind the interface chemistry that drives the observed process. Further it is (from the main text) not even clear to me if there is a full transformation to a complete dispersed system and what the final drop size distribution and the stability of the emulsion really are.

I fully agree with the authors that this topic is highly interesting and certainly publishable, but I also believe that they have not provided enough information to fully capture the nature of the emulsification process. I was also wondering if these effects are universal or only work with the

specific system using DCE as the oil phase. Their TOC graphic seems to imply that this is universal but I could not see that there is evidence for it. In this early stage of the work I cannot support publication in Nature Comm.

Response to Reviewer Comments:

Reviewer #1 (Remarks to the Author):

In this manuscript, the authors report spontaneous emulsification at the water/oil (1,2-dichloroethane, DCE) interface caused by the anion flux between the two phases. The interfacial flux of anions is enhanced by a phase-transfer agent (NBu_4^+) present in the oil phase. The emulsification can be tuned by tuning the concentration and type of anions in the two phases. The emulsification mechanism was well discussed. The research topic is interesting and the results are reasonable. I recommend this manuscript to be published after minor revision. Specific comments:

1. Page 3, for the partitioning test of chloroauric acid between the water and oil phases (Figure S1), KCl was used as the supporting electrolyte in the aqueous phase. Do KCl and NaCl have different effects on emulsification?

We thank the reviewer for pointing this out. We have repeated emulsification experiments with KCl and see no observable difference in behavior. We have added the microscope images of the liquid|liquid boundary for this case when 1 M KCl was used instead of 1 M NaCl in Figure S8 and have reproduced them below for convenience. Both images below were conducted with 10 mM HAuCl_4 (aq) and 0.1 M $[\text{NBu}_4][\text{ClO}_4]$, but part A of the image below was conducted with 1 M KCl added to the water phase, whereas part B used 1 M NaCl added to the water phase.

Figure S8: Experimental control for the effect of NaCl vs. KCl on emulsification behavior. The DCE phase contained 0.1 M $[\text{NBu}_4][\text{ClO}_4]$ and the aqueous phase contained 10 mM HAuCl_4 and 1 M KCl (A) or 1 M NaCl (B). A microemulsion still formed at the liquid|liquid boundary, indicating that using NaCl vs. KCl in the aqueous phase has little effect on the spontaneous emulsification. These images were taken with a 40x NA 0.60 objective and a 500 ms exposure time.

We have also included the following sentence in the main text for clarity:

“Secondly, changing the added aqueous electrolyte from 1 M NaCl to 1 M KCl shows identical results (see Figure S8), indicating that changing the supporting electrolyte has no observable effect.”

2. Figure S5, it seems that many droplets exist in the DCE phase. Can water droplets form in the oil phase when no NaCl is present in the water phase?

We thank the reviewer for the opportunity to clarify this point. Water droplets do indeed form in the oil phase for the chloroauric acid system when no salt is present. This point highlights the importance of ionic strength: when salt is present, it will compete for hydration, diminishing the chance that water droplets will form. We have added the following sentences to the main text to clarify this point:

“For fluxification with chloroauric acid, water droplets can be observed in the DCE phase in the absence of additional electrolyte in the water phase. This observation highlights the importance of ion hydration during the partitioning: when other salt molecules in a far greater abundance compete for hydration during ion transfer, the probability of water droplet formation in the oil phase diminishes.”

For example, Figure S11 (shown below) demonstrates the role of NaCl for multiple organic solvents:

Figure S11: Light microscopy images taken at the water|DCE boundary when the aqueous phase contained 10 mM HAuCl₄ and the oil phase contained 0.1 M [NBu₄][ClO₄] for (A) dichloromethane, (B) nitrobenzene, and (C) chloroform with either 1 M NaCl (i) or 0 M NaCl (ii) present in the aqueous phase. The water and DCE phases are indicated, and the scale bars are

25 μm . All optical micrographs were taken with a 40x NA 0.60 objective and a 500 ms exposure time.

3. Page 10, line 200, “ ClO_4^- (B), Br^- (C)” is incorrect?

We thank the reviewer for identifying this error. The figure caption is now correctly labeled as “ Br^- (B), ClO_4^- (C)”

4. Page 12, line 222, it is better to write “the number of emulsion droplets” instead of “the number of microemulsion droplets”.

Thank you, this has been adjusted in the main text.

“... Histograms show the number of emulsion droplets measured (B) and the average area of measured droplets (C) as a function of distance from the liquid|liquid interface. All droplets were measured after 10 minutes from the initial time of contact.”

Reviewer #2 (Remarks to the Author):

This manuscript by Colón-Quintana et al. demonstrates a method of emulsification at the oil/water interface using solute flux enhanced by a phase transfer reagent. They achieve emulsification using tetrachloroaurate as the solute in the aqueous phase and tetrabutylammonium cation as the phase transfer agent in the oil phase. When the aqueous phase containing HAuCl_4 comes into contact with an oil phase containing $[\text{NBu}_4][\text{ClO}_4]$, $[\text{NBu}_4]^+$ ions facilitate the transfer of AuCl_4^- ions into the oil phase. To maintain electroneutrality in the aqueous phase, ClO_4^- anions are transferred into the aqueous phase. The degree of emulsification was found to be directly correlated to ion concentrations. The emulsification process was also sensitive to the hydrophobicity of cations and anions in the oil phase.

Utilizing solute flux as the driving force for emulsification is certainly interesting. Emulsification is characterized here using optical microscopy, and I found the optical microscopy images quite clear and easy to follow. The experiments that show the effect of the identity of organic phase ions are also nice and convincing. However, in its current form, the manuscript does not provide a clear understanding of flux-induced emulsification. Moreover, the authors have not provided enough background for the reader to understand their choice of reagents and their mechanistic role in the emulsification process. If both of these concerns are addressed, I believe the results in the manuscript can be significant for a wider audience. In the following, I list my specific questions and suggestions:

1. The criteria for the choice of AuCl_4^- as the aqueous phase anion needs to be stated. If flux-induced emulsification is a general phenomenon, what are the relevant parameters to consider when choosing the aqueous phase anion? If the aqueous phase contained other large, weakly hydrated anions such as SCN^- , would they also give rise to emulsification?

We are grateful to this reviewer for bringing up these points, as they have launched us on an exciting journey into a detailed understanding of our system. The reviewer is correct in thinking that weakly hydrated anions will also cause spontaneous emulsification. We now present compelling results that flux-induced emulsification occurs when KSCN is used in the aqueous phase. We have also previously shown that in the presence of tetrabutylammonium, hexacyanoferrate (III), a trivalent anion, will partition into the DCE phase.¹ We have added the resulting Figure 5 and a microscopic model to the main text and have reproduced it below for convenience:

Figure 5: Light microscopy images taken at the water|DCE boundary when the aqueous phase contained 10 mM HAuCl₄ (A), KSCN (B), or K₃[Fe(CN)₆] (C) with either 1 M KCl (i) or 0 M KCl (ii). The DCE phase always contained 0.1 M [NBu₄][ClO₄]. The water and DCE phases are indicated, and the scale bars are 25 μm. (D) Proposed microscopic model of how the transferring ions can cause water-in-oil droplets and/or oil-in-water droplets to form and be stabilized by antagonistic salts. All optical micrographs were taken with a 40x NA 0.60 objective and a 500 ms exposure time.

We have further shown that flux-induced emulsification is conserved not only with DCE but with chloroform, dichloromethane, and nitrobenzene. These results are given in Figure S11 and reproduced below for convenience:

Figure S11: Light microscopy images taken at the water|DCE boundary when the aqueous phase contained 10 mM HAuCl_4 and the oil phase contained 0.1 M $[\text{NBu}_4][\text{ClO}_4]$ for (A) dichloromethane, (B) nitrobenzene, and (C) chloroform with either 1 M NaCl (i) or 0 M NaCl (ii) present in the aqueous phase. The water and DCE phases are indicated, and the scale bars are 25 μm . All optical micrographs were taken with a 40x NA 0.60 objective and a 500 ms exposure time.”

2. Similarly, how are the required concentrations of the aqueous phase and organic phase ions and the aqueous electrolyte (1 M) decided? To elaborate this question further, The concentration series in Figure 2 uses 0,2.5, 5, and 7.5 mM of HAuCl_4 and 0,10, 25, and 50 mM $[\text{NBu}_4][\text{ClO}_4]$ – why not use a 1:1 ratio of cations and anions? How were the concentrations selected? Could authors also comment on the role of 1M NaCl in this process?

We appreciate the opportunity to clarify why we decided on the concentrations we chose. Because we are using a phase transfer agent (i.e, the tetrabutylammonium cation used for this Figure’s results) to drive anionic flux, we chose to keep this concentration in excess for most of the studies. However, the point of Figure 2 was to demonstrate that if the initial concentrations of the reactants in Equation (1) are modified, we see a corresponding change in the degree of emulsification. Especially with the chloroauric acid results, using excess concentrations of the phase transfer agent can reduce the amount of relatively more expensive chloroauric acid to

achieve sufficient ion flux. We have also presented results from the reviewer's proposed experiment with a 1:1 ratio of cations and anions. These results are shown in Figure 2B(ii) and Figure 3 and still show emulsification, but to a lesser extent than the traditional experiments where excess $[\text{NBu}_4][\text{ClO}_4]$ is used (e.g., 10 mM HAuCl_4 and 100 mM $[\text{NBu}_4][\text{ClO}_4]$).

The reviewer brings up a great point about the role of NaCl. Removing NaCl from the system allows droplets to form on both sides of the interface. This behavior was observed for multiple oil phases as shown in Figure S11 as shown above. We have added the following sentences to the main text:

“From Figure S9, we have found that the presence of additional aqueous electrolyte (NaCl or KCl) does not seem to influence the DCE droplet formation in the aqueous phase but does influence whether water droplets can form in the DCE phase. This can be observed as well in Figure 5A. To explain this case, it is important to consider the solvation of the transferring aqueous anion and that transferring ions bring water molecules with them. For example, molecular dynamics simulations have shown that ions transferring across liquid/liquid boundaries bring solvent molecules with them.^{21,22,23} When 1 M KCl is present in the water, the AuCl_4^- will become less solvated in the aqueous phase and will transfer with less solvent molecules because of solvation competition with other ions in greater excess. When KCl or NaCl is absent in the aqueous phase, more water molecules can transfer with the anions such that water droplets can form in the DCE. This is also the case when other aqueous anions (SCN^- and $[\text{Fe}(\text{CN})_6]^{3-}$) transfer into the DCE phase. Interestingly, DCE droplets do not form in the aqueous phase when using KSCN or $\text{K}_3[\text{Fe}(\text{CN})_6]$ (Figure 5B-C). This is likely because DCE droplets can be stabilized in some cases by an antagonistic salt (as is the case when using HAuCl_4).²⁴ This is represented in our proposed microscopic mechanism shown in Figure 5D.”

3. It needs to be made clear how the droplets are formed and stabilized here. For example, if we compare this method to a more well-known emulsification process, such as surfactant-based emulsification, how are these droplet interfaces different? Do the cations and anions both occupy the surface of the droplets in a double-layer fashion? The droplets are always formed on the aqueous side: How are the droplets transferred to that side?

We now propose a microscopic model in light of all of our experiments, controls, and generalizations. When the complex ions form at the interface, they may form antagonistic salts. Such salts have been shown to stabilize curved interfaces. For instance, tetrabutylammonium chloroaurate is likely an antagonistic salt that, relative to other species in solution, has a high affinity for the liquid/liquid interface. We contend that the flux perturbs the boundary and active ion transfer creates solvent fingers at the boundary that can pinch off and become droplets in the presence of a stabilizer. We think of this process as a meandering river that creates an oxbow lake. We have reproduced part of the section on Generalizability and Microscopic Model below for convenience:

“Generalizability and Microscopic Model:

In this section, we use lessons learned from the chloroauric acid system described above to develop a more generalized platform for emulsification. Partitioning ions between the water and oil phase follow an equilibrium reaction like that of Equation 1, but other anions can be used instead of AuCl_4^- . In addition to using HAuCl_4 in the aqueous phase, we also used KSCN and $\text{K}_3[\text{Fe}(\text{CN})_6]$ as a source of different partitioning anions to induce emulsification (Figure 5). Tetraalkylammonium cations have been shown to transfer thiocyanate ions from water to oil,¹⁵ and ferricyanide has similarly been shown to transfer to the DCE phase in the presence of $[\text{NBu}_4][\text{ClO}_4]$.²⁰

Figure 5: Light microscopy images taken at the water|DCE boundary when the aqueous phase contained 10 mM HAuCl₄ (A), KSCN (B), or K₃[Fe(CN)₆] (C) with either 1 M KCl (i) or 0 M KCl (ii). The DCE phase always contained 0.1 M [NBu₄][ClO₄]. The water and DCE phases are indicated, and the scale bars are 25 μm. (D) Proposed microscopic model of how the transferring ions can cause water-in-oil droplets and/or oil-in-water droplets to form and be stabilized by antagonistic salts. All optical micrographs were taken with a 40x NA 0.60 objective and a 500 ms exposure time.

From Figure S9, we have found that the presence of additional aqueous electrolyte (NaCl or KCl) does not seem to influence the DCE droplet formation in the aqueous phase but does influence whether water droplets can form in the DCE phase. This can be observed as well in Figure 5A. To explain this case, it is important to consider the solvation of the transferring aqueous anion and that transferring ions bring water molecules with them. For example, molecular dynamics simulations have shown that ions transferring across liquid|liquid boundaries bring solvent molecules with them.^{21,22,23} When 1 M KCl is present in the water, the AuCl_4^- will become less solvated in the aqueous phase and will transfer with less solvent molecules because of solvation competition with other ions in greater excess. When KCl or NaCl is absent in the aqueous phase, more water molecules can transfer with the anions such that water droplets can form in the DCE. This is also the case when other aqueous anions (SCN^- and $[\text{Fe}(\text{CN})_6]^{3-}$) transfer into the DCE phase. Interestingly, DCE droplets do not form in the aqueous phase when using KSCN or $\text{K}_3[\text{Fe}(\text{CN})_6]$ (Figure 5B-C). This is likely because DCE droplets can be stabilized in some cases by an antagonistic salt (as is the case when using HAuCl_4).²⁴ This is represented in our proposed microscopic mechanism shown in Figure 5D.

Furthermore, to show generalizability, we used HAuCl_4 and $[\text{NBu}_4][\text{ClO}_4]$ to induce emulsification for various water|oil interfaces. While all observations thus far have been with water and DCE, we also dissolved 0.1 M $[\text{NBu}_4][\text{ClO}_4]$ into dichloromethane, chloroform, and nitrobenzene. Images of the interface of these solutions in contact with aqueous solutions of 10 mM HAuCl_4 show similar behavior to the results with DCE for cases both with and without added NaCl in the aqueous phase (Figure S11).

All of our observations to this point allow us to propose a generalized microscopic model for the flux-induced emulsification, as shown in Figure 5D. While the emulsification is driven by partitioning of ions at the boundary and flux of ions to the boundary, the mechanism of ion transfer is more involved. Our results suggest that when ions transfer across the interface, complex ions can form that can stabilize curved interfaces. Such ions are called antagonistic salts.²⁴ In the case of chloroauric acid in water, tetrabutylammonium chloroaurate forms at the liquid|liquid interface. In a complex matrix, molecules that have the highest affinity for the boundary will adsorb to the boundary. Thus, we expect droplets formed in either phase to be stabilized by the same antagonistic salt. The sinuosity of the liquid|liquid interface has been shown, and transferring ions increase the surface roughness of this interface, namely via “water fingers”.²⁵ Ions that transfer from water to oil have been shown to bring water molecules with them, forming a finger-like structure; whereas the transfer of oil-phase ions into water has not been shown to bring in solvent molecules and instead is facilitated by water fingers that can engulf oil-phase ions.²¹ Thus, substantial ion-transfer can create morphologies like that shown in the middle panels in Figure 5D due to many water fingers. Protrusions of water fingers can pinch off upon further ionic flux in all cases. However, under cases where there is a sufficiently high degree of ion flux, the density of water fingers is high enough such that water can enclose regions of oil to form oil droplets. This is likely why AuCl_4^- , which strongly partitions into the DCE phase, creates DCE droplets in water (Figure 5A), whereas other ions with lower partition coefficients, like SCN^- and $[\text{Fe}(\text{CN})_6]^{3-}$, cannot form dense enough regions of water fingers to enclose oil droplets and instead only produce water droplets (Figures 5B-C). We have measured the partition coefficient of SCN^- from water to DCE to be 0.07 via a colorimetric experiment (Figure S12) and the partition coefficient of $[\text{Fe}(\text{CN})_6]^{3-}$ to be 0.03 via a voltammetric experiment (Figure S13). These partition coefficients are orders of magnitude

smaller than the corresponding partition coefficient for AuCl_4^- , which explains the lack of DCE droplets formed in these cases. Our model is reminiscent of a meandering river that creates oxbow lakes over time.²⁶

While our model predicts droplets form in both phases, the fate of those droplets depends strongly on how well the antagonistic salt stabilizes a droplet in a given phase. For our results with KSCN and $\text{K}_3[\text{Fe}(\text{CN})_6]$, no DCE droplets are observed in the aqueous phase. Our model predicts that tetrabutylammonium thiocyanate is not an effective stabilizer of DCE droplets in water. Furthermore, $\text{Fe}(\text{CN})_6^{3-}$ is the most hydrophilic example where one can observe flux-induced emulsification. Thus, another important aspect of our model is the solvation about the transferring ions. Ferricyanide will be much more hydrated, which can increase the probability of water droplet formation.

Given our model and observations, we offer the following design principles for flux-induced emulsification (fluxification):

- 1.) Only systems that are amenable to a phase transfer agent will allow for emulsification.
- 2.) The extent of droplet formation depends on the flux at the liquid|liquid interface: the higher the flux, the more quickly droplets will form. From Figure 3, faster flux produces smaller droplets likely because droplets have not had enough time to coalesce.
- 3.) The probability of droplet formation depends on the following parameters:
 - a. Hydrophilicity/hydrophobicity and hardness/softness of the transferring ions.
 - b. Solvation competition with other ions in solution.
 - c. Ability of complex ions (antagonistic salts) to stabilize droplets in one phase over another. The orientation of the antagonistic salts at the boundary must also play a role (not considered in this work).

Given the rising interest in studying and engineering chemistry in microenvironments,^{27,28,29} our system offers a valuable new way to emulsify a water and oil phase without the use of large amounts of amphiphilic solvents or additional organic solutes. This system uses a phase-transfer agent to drastically increase the flux of an anion (e.g., AuCl_4^-) and the degree of emulsification can be tuned by the hydrophilicity and/or concentration of a secondary anion in the DCE solution. Furthermore, we can get water droplets in the oil phase or oil droplets in the aqueous phase by adding or removing additional electrolyte in the water phase. The ability to tune the extent of emulsification with the choice of electrolyte provides a robust alternative to spontaneous emulsification methods via more complex strategies. Finally, our results offer insight into a new physical means by which nature can spontaneously produce microcompartments, a topic of intense interest in understanding the abiotic origins of life.^{27, 30,31,32} ”

4. How long are the droplets stable? Some information on the time stability of these emulsions needs to be added.

We have now performed Dynamic Light Scattering (DLS), and the results are reported in the Supporting Information Figure S4-6 and reproduced immediately below for convenience. We chose to only take results from DLS where the polydispersity index was below 0.5 (where 0.1 is considered quite monodisperse). To perform this experiment, we chose the chloroaurate system because it was most efficient at creating droplets. DLS measurements showed a monomodal distribution of droplets on the order of 100s of nm to a few micrometers that were stable for up

to 3 hours. The solution also appeared cloudy for several hours. We have included the relevant SI figures below for the reviewer's convenience:

Figure S4: Dynamic light scattering measurements of DCE droplets spontaneously formed in the aqueous phase when 10 mL HAuCl₄ was pipetted into a 10 mL of DCE containing 0.1 M [NBu₄][ClO₄]. An overhead stirrer was inserted into the aqueous phase and was used to induce convection in the aqueous phase without making contact with the liquid|liquid boundary. Droplets were collected after ten minutes of solution contact and measured for over an hour. Monomodal distributions are observed for all cases with the droplet diameters decreasing with time.

Figure S5: Images for the stability of the observed emulsion over time. A 10 mM HAuCl₄ aqueous solution was pipetted over a 0.1 M [NBu₄][ClO₄] DCE solution. The emulsion was allowed to form spontaneously at $t = 0$ min, afterwards overhead stirring was used for 1 min to induce convection and maximize emulsion formation. The observed emulsion was then

monitored over time at A) 0 min, B) 2 min, C) 203 min, D) 261 min, and E) 324 min. A noticeable emulsion can be seen up to 261 min showing stability over long periods of time.

We have added the following discussion to the manuscript to communicate this to the reader:

“To characterize the stability and reproducibility of the observed droplets, dynamic light scattering (DLS) measurements were performed on the droplets that are spontaneously formed. To achieve this, 10 mL of 10 mM H_{AuCl}₄ was pipetted over 10 mL of DCE containing 0.1 M [NBu₄][ClO₄]. An overhead stirrer was inserted into the aqueous phase and was used to induce convection in the aqueous phase without making contact with the liquid/liquid boundary. This allowed the spontaneously formed DCE droplets to be suspended in the aqueous phase. DLS measurements were performed on this suspension for over an hour, and the droplet sizes are reported in Figure S4. These results show that the droplets are stable for over an hour, with monomodal distributions decreasing in diameter from just over 1 μm to less than 1 μm within the hour. Additionally, images were taken of the two bulk phases during flux-induced emulsification without added convection for several hours and the aqueous phase appeared cloudy due to the emulsion for a few hours after initial contact (see Figure S5). These results suggest that these droplets can be stable for long periods of time.”

5. Figure 2 shows the effect of the anions and cations' concentration dependence on emulsification efficiency. To quantify emulsification efficiency, authors have used histograms of the number of droplets versus distance from the liquid/liquid interface. However, these plots do not show a proportional emulsification increase with increasing ionic concentration. For example, according to Figure 2E, the total number of droplets is higher at 5 mM H_{AuCl}₄ than 10 mM H_{AuCl}₄, which contradicts the authors' conclusion. Moreover, it is not clear if this type of histogram is the appropriate way of analysis here. A plot of the total number of droplets as a function of concentration would be more informative.

We apologize for any possible misconceptions. The reviewer is correct in that a greater number of droplets are observed in the case of 5 mM H_{AuCl}₄ as compared to 10 mM H_{AuCl}₄. However, if one looks at the average cross-sectional area measurements (Figure 2F), the 5 mM case has the smallest droplet sizes at all but one of the distance ranges from the interface (and certainly smaller than the 10 mM case). Therefore we find that it is important to not just report the total number of droplets, but also the average droplet size of the formed droplets.

In an attempt to be clearer, we have added the following sentences to the manuscript:

“Examining the number of droplets identified at different concentrations does not provide a clear summary of the emulsification (for example, a greater total number of droplets were identified for the case where [H_{AuCl}₄] = 5 mM as compared to other concentrations of H_{AuCl}₄). However, examining the cross-sectional areas sheds light on this observation. The average cross-sectional area of droplets were the largest under the conditions of the highest concentrations of H_{AuCl}₄ or the [NBu₄][ClO₄] (Figures 2F and 2H). So despite forming a greater number of droplets when [H_{AuCl}₄] = 5 mM, a majority of these droplets are smaller than the droplets at other concentrations, as indicated by the

average cross-sectional areas. Due to the fact that droplets coalesce, analyzing the droplet counts alone is insufficient to characterize the extent of emulsification.”

6. Figure 2F & H: what do the error bars represent?

We thank the reviewer for pointing out this missing information. We have now indicated what the error bars represent within the figure caption, as shown below:

“**Figure 2:** A) Optical micrographs of the effect of HAuCl_4 concentration on droplet formation. An increase in concentration can be seen sequentially for I (0 mM), II (2.5 mM), III (5 mM), and IV (7.5 mM). B) Optical micrographs of the effect of $[\text{NBu}_4][\text{ClO}_4]$ concentration on droplet formation. An increase in concentration can be seen sequentially for I (0 mM), II (10 mM), III (25 mM), and IV (50 mM). C) Optical micrograph for the droplet formation at 10 mM HAuCl_4 and 100 mM $[\text{NBu}_4][\text{ClO}_4]$. D) Schematic representation of the proposed mechanism for partitioning of chloroaurate from aqueous to organic media in the presence of NBu_4^+ while maintaining electroneutrality with ClO_4^- . Histograms presented below schematic show the effect of changes in concentration on the frequency (E, G) and average cross-sectional area (F, H) of droplets as a function of distance from the interface for chloroauric acid (Blue; E and F) and tetrabutylammonium perchlorate (Red; G and H) respectfully. The error bars in F and H correspond to standard deviations about the mean for N equal to the number of droplets identified in that region (from E and G). Data for histograms was gathered 10 minutes after initial contact of aqueous and DCE phases. All optical micrographs were taken with a 40x NA 0.60 objective and a 500 ms exposure time. ”

7. Some details on the emulsification process would be useful for reproducing the results: How reproducible are the droplet counts as a function of concentration? (Can the authors provide error bars on these histograms?) Does the order of mixing of the aqueous and organic phases influence the result?

We have provided better characterization of droplet sizes with dynamic light scattering (DLS) and have shown consistent droplet diameters (Figure S6). We have now included statistical analysis for droplet size and reproducibility measurements using dynamic light scattering including: the average size, standard deviation, and relative standard deviation for multiple samples and for a combined average size. As shown below:

Figure S6: Droplet size reproducibility measurements using dynamic light scattering. DCE droplets were spontaneously formed in the aqueous phase when 2 mL of 10 mM H₂AuCl₄ was pipetted above 2 mL of DCE containing 0.1 M [NBu₄][ClO₄]. All droplet dynamic light scattering measurements were taken 5 minutes after contact of both solutions for an N = 4.

	Sample 1	Sample 2	Sample 3	Sample 4	Combined Avg. Size
Average Size (nm)	1230.5	1441.0	1611.8	1283.9	1391.8
Std. Dev. (nm)	135.30	157.11	182.95	187.08	171.72
RSD (%)	10.995	10.902	11.350	14.571	12.337

Table S2: Data and statistical analysis for droplet size and reproducibility measurements using dynamic light scattering (Figure S6). The average size, standard deviation, and relative standard deviation are provided for all samples measured and for a combined average size.

The fact that the distribution remains monomodal at similar diameters implicates that the droplet formation process is rather consistent for a given set of conditions (e.g., transferring ion, ion concentrations, type of oil phase, etc.).

8. On page 4, line 80, the authors state, "Based on ion transfer potentials, the transfer of perchlorate ion from the DCE phase to the aqueous phase would be the most likely process (as opposed to proton or potassium ions transferring from the aqueous phase to the DCE). Can the authors provide the ion transfer potential values of all the cations and anions involved (including the supporting electrolyte) to support this argument? A table with all these values will make it much easier for the reader to understand the claim.

We thank the reviewer for this request. We have now provided the standard Gibbs free energy changes and ion transfer potentials in the supporting information file (Table S1) and agree that this helps the reader understand the claims of this report.

Ion:	$\Delta G_{w \rightarrow DCE}^0$ (kJ/mol)	$\Delta \phi_{w \rightarrow DCE}^0$ (mV)
H ⁺	60.5 ⁽¹⁾	-627
Na ⁺	58.9 ⁽¹⁾	-610
K ⁺	52.9 ⁽¹⁾	-548
NEt ₄ ⁺	4.2 ⁽²⁾	-44
NBu ₄ ⁺	-21.8 ⁽²⁾	226
Cl ⁻	-46.4 ⁽²⁾	481
ClO ₄ ⁻	-17.2 ⁽²⁾	178
Br ⁻	-38.5 ⁽²⁾	399
PF ₆ ⁻	8.7 ⁽¹⁾	-90

Table S1: The standard Gibbs free energy of ion transfer from water (w) to DCE, as reported in citations (1) and (2), and the standard ion transfer potentials of these ions from water to DCE.

9. On page 6, line 111, "At all-time points, the average droplet area decays with decreasing distance from the interface." This should be "increasing distance"

We thank the reviewer for identifying this error. This sentence has been correctly changed to:

"At all time points, the average droplet area decays with increasing distance from the interface."

10. Schematic 1 shows 1M NaCl in the aqueous phase, whereas the text (page 3, line 71) says it is 1M KCl – this needs to be corrected.

After examination, the original text was correct. We did use 1 M KCl for the experiment shown in Figure S1, but we used 1 M NaCl for most of the other original experiments shown in the manuscript. However, we have performed additional experiments that show that there is no appreciable difference in emulsification between using 1M KCl or 1 M NaCl (see Figure S8). We have been explicit throughout the text throughout whether we are using KCl or NaCl and have added the following sentence to clarify:

"Secondly, changing the added aqueous electrolyte from 1 M NaCl to 1 M KCl shows identical results (see Figure S8), indicating that changing the supporting electrolyte has a minimal effect."

Figure S8: Experimental control for the effect of NaCl vs. KCl on emulsification behavior. The DCE phase contained 0.1 M $[\text{NBu}_4][\text{ClO}_4]$ and the aqueous phase contained 10 mM HAuCl_4 and 1 M KCl (A) or 1 M NaCl (B). A microemulsion still formed at the liquid|liquid boundary, indicating that using NaCl vs. KCl in the aqueous phase has little effect on the spontaneous emulsification. These images were taken with a 40x NA 0.60 objective and a 500 ms exposure time.

11. The experiments by varying the hydrophobicity of the organic phase anion and cations are nicely done. Can the authors provide general guidelines for choosing the appropriate ionic species (both in the aqueous and organic phase) for a given water/oil interface (perhaps based on oil/water partition coefficients or solvation free energies?). Adding such a section would make the manuscript relevant for a wider audience.

We have now added additional experiments using various partitioning salts (including HAuCl_4 , KSCN , and $\text{K}_3[\text{Fe}(\text{CN})_6]$) as well as various oil phases (1,2-dichloroethane, dichloromethane, nitrobenzene, and chloroform). These results are summarized in Figure 5 and Figure S11. Interestingly we have shown how to control whether we get DCE droplets in water or water droplets in DCE (or both at the same time). We have also shown how to control the degree of emulsification (by changing the nature and/or concentrations of the ions). This has been reflected in the discussion in the main text in several places and is reproduced below for convenience.

“Generalizability and Microscopic Model:

In this section, we use lessons learned from the chloroauric acid system described above to develop a more generalized platform for emulsification. Partitioning ions between the water and oil phase follow an equilibrium reaction like that of Equation 1, but other anions can be used instead of AuCl_4^- . In addition to using HAuCl_4 in the aqueous phase, we also used KSCN and $\text{K}_3[\text{Fe}(\text{CN})_6]$ as a source of different partitioning anions to induce emulsification (Figure 5). Tetraalkylammonium cations have been shown to transfer thiocyanate ions from water to oil,¹⁵ and

ferricyanide has similarly been shown to transfer to the DCE phase in the presence of $[\text{NBu}_4][\text{ClO}_4]$.²⁰

Figure 5: Light microscopy images taken at the water|DCE boundary when the aqueous phase contained 10 mM HAuCl_4 (A), KSCN (B), or $\text{K}_3[\text{Fe}(\text{CN})_6]$ (C) with either 1 M KCl (i) or 0 M KCl (ii). The DCE phase always contained 0.1 M $[\text{NBu}_4][\text{ClO}_4]$. The water and DCE phases are indicated, and the scale bars are 25 μm . (D) Proposed microscopic model of how the transferring ions can cause water-in-oil droplets and/or oil-in-water droplets to form and be stabilized by

antagonistic salts. All optical micrographs were taken with a 40x NA 0.60 objective and a 500 ms exposure time.

From Figure S9, we have found that the presence of additional aqueous electrolyte (NaCl or KCl) does not seem to influence the DCE droplet formation in the aqueous phase but does influence whether water droplets can form in the DCE phase. This can be observed as well in Figure 5A. To explain this case, it is important to consider the solvation of the transferring aqueous anion and that transferring ions bring water molecules with them. For example, molecular dynamics simulations have shown that ions transferring across liquid|liquid boundaries bring solvent molecules with them.^{21,22,23} When 1 M KCl is present in the water, the AuCl_4^- will become less solvated in the aqueous phase and will transfer with less solvent molecules because of solvation competition with other ions in greater excess. When KCl or NaCl is absent in the aqueous phase, more water molecules can transfer with the anions such that water droplets can form in the DCE. This is also the case when other aqueous anions (SCN^- and $[\text{Fe}(\text{CN})_6]^{3-}$) transfer into the DCE phase. Interestingly, DCE droplets do not form in the aqueous phase when using KSCN or $\text{K}_3[\text{Fe}(\text{CN})_6]$ (Figure 5B-C). This is likely because DCE droplets can be stabilized in some cases by an antagonistic salt (as is the case when using HAuCl_4).²⁴ This is represented in our proposed microscopic mechanism shown in Figure 5D.

Furthermore, to show generalizability, we used HAuCl_4 and $[\text{NBu}_4][\text{ClO}_4]$ to induce emulsification for various water|oil interfaces. While all observations thus far have been with water and DCE, we also dissolved 0.1 M $[\text{NBu}_4][\text{ClO}_4]$ into dichloromethane, chloroform, and nitrobenzene. Images of the interface of these solutions in contact with aqueous solutions of 10 mM HAuCl_4 show similar behavior to the results with DCE for cases both with and without added NaCl in the aqueous phase (Figure S11).

All of our observations to this point allow us to propose a generalized microscopic model for the flux-induced emulsification, as shown in Figure 5D. While the emulsification is driven by partitioning of ions at the boundary and flux of ions to the boundary, the mechanism of ion transfer is more involved. Our results suggest that when ions transfer across the interface, complex ions can form that can stabilize curved interfaces. Such ions are called antagonistic salts.²⁴ In the case of chloroauric acid in water, tetrabutylammonium chloroaurate forms at the liquid|liquid interface. In a complex matrix, molecules that have the highest affinity for the boundary will adsorb to the boundary. Thus, we expect droplets formed in either phase to be stabilized by the same antagonistic salt. The sinuosity of the liquid|liquid interface has been shown, and transferring ions increase the surface roughness of this interface, namely via “water fingers”.²⁵ Ions that transfer from water to oil have been shown to bring water molecules with them, forming a finger-like structure; whereas the transfer of oil-phase ions into water has not been shown to bring in solvent molecules and instead is facilitated by water fingers that can engulf oil-phase ions.²¹ Thus, substantial ion-transfer can create morphologies like that shown in the middle panels in Figure 5D due to many water fingers. Protrusions of water fingers can pinch off upon further ionic flux in all cases. However, under cases where there is a sufficiently high degree of ion flux, the density of water fingers is high enough such that water can enclose regions of oil to form oil droplets. This is likely why AuCl_4^- , which strongly partitions into the DCE phase, creates DCE droplets in water (Figure 5A), whereas other ions with lower partition coefficients, like SCN^- and $[\text{Fe}(\text{CN})_6]^{3-}$, cannot form dense enough regions of water fingers to enclose oil droplets and instead only produce water droplets (Figures 5B-C). We have measured the partition coefficient of SCN^- from water to DCE to be 0.07

via a colorimetric experiment (Figure S12) and the partition coefficient of $\text{Fe}(\text{CN})_6^{3-}$ to be 0.03 via a voltammetric experiment (Figure S13). These partition coefficients are orders of magnitude smaller than the corresponding partition coefficient for AuCl_4^- , which explains the lack of DCE droplets formed in these cases. Our model is reminiscent of a meandering river that creates oxbow lakes over time.²⁶

While our model predicts droplets form in both phases, the fate of those droplets depends strongly on how well the antagonistic salt stabilizes a droplet in a given phase. For our results with KSCN and $\text{K}_3[\text{Fe}(\text{CN})_6]$, no DCE droplets are observed in the aqueous phase. Our model predicts that tetrabutylammonium thiocyanate is not an effective stabilizer of DCE droplets in water. Furthermore, $\text{Fe}(\text{CN})_6^{3-}$ is the most hydrophilic example where one can observe flux-induced emulsification. Thus, another important aspect of our model is the solvation about the transferring ions. Ferricyanide will be much more hydrated, which can increase the probability of water droplet formation.

Given our model and observations, we offer the following design principles for flux-induced emulsification (fluxification):

- 4.) Only systems that are amenable to a phase transfer agent will allow for emulsification.
- 5.) The extent of droplet formation depends on the flux at the liquid|liquid interface: the higher the flux, the more quickly droplets will form. From Figure 3, faster flux produces smaller droplets likely because droplets have not had enough time to coalesce.
- 6.) The probability of droplet formation depends on the following parameters:
 - a. Hydrophilicity/hydrophobicity and hardness/softness of the transferring ions.
 - b. Solvation competition with other ions in solution.
 - c. Ability of complex ions (antagonistic salts) to stabilize droplets in one phase over another. The orientation of the antagonistic salts at the boundary must also play a role (not considered in this work).

Given the rising interest in studying and engineering chemistry in microenvironments,^{27,28,29} our system offers a valuable new way to emulsify a water and oil phase without the use of large amounts of amphiphilic solvents or additional organic solutes. This system uses a phase-transfer agent to drastically increase the flux of an anion (e.g., AuCl_4^-) and the degree of emulsification can be tuned by the hydrophilicity and/or concentration of a secondary anion in the DCE solution. Furthermore, we can get water droplets in the oil phase or oil droplets in the aqueous phase by adding or removing additional electrolyte in the water phase. The ability to tune the extent of emulsification with the choice of electrolyte provides a robust alternative to spontaneous emulsification methods via more complex strategies. Finally, our results offer insight into a new physical means by which nature can spontaneously produce microcompartments, a topic of intense interest in understanding the abiotic origins of life.^{27, 30,31,32} »

Reviewer #3 (Remarks to the Author):

Colón-Quintana et al. report in their study on spontaneous formation of emulsion using AuCl_4^- as a phase transfer agent. The authors provide a parameter study where they varied the concentration of the “emulsifier” and performed some control experiments. emulsification was measured by recording light microscopy images close to the oil-water interface.

I believe this study provides information on an intriguing phenomenon that is highly interesting from a fundamental aspect. The authors experiments helped to show that AuCl_4^- is important for the self-emulsification. However, they could not reveal the molecular mechanism behind the interface chemistry that drives the observed process. Further it is (from the main text) not even clear to me if there is a full transformation to a complete dispersed system and what the final drop size distribution and the stability of the emulsion really are.

I fully agree with the authors that this topic is highly interesting and certainly publishable, but I also believe that they have not provided enough information to fully capture the nature of the emulsification process. I was also wondering if these effects are universal or only work with the specific system using DCE as the oil phase. Their TOC graphic seems to imply that this is universal but I could not see that there is evidence for it. In this early stage of the work I cannot support publication in Nature Comm.

We are grateful to the reviewer for taking the time to review our manuscript. We have worked very hard to address the reviewer’s specific points, namely: mechanism, size distribution, stability, and universality. We have studied size distribution and stability using Dynamic Light Scattering. We have demonstrated universality in two ways: 1.) by performing the experiment in different oils (dichloromethane, chloroform, and nitrobenzene.), and 2.) by showing other systems amenable to phase transfer (potassium hexacyanoferrate (III) and potassium thiocyanate) also display flux-induced emulsification. Finally, we propose a microscopic mechanism given all of our results. The new data are provided below for convenience.

Figure S4: Dynamic light scattering measurements of DCE droplets spontaneously formed in the aqueous phase when 10 mL HAuCl_4 was pipetted into a 10 mL of DCE containing 0.1 M $[\text{NBu}_4][\text{ClO}_4]$. An overhead stirrer was inserted into the aqueous phase and was used to induce convection in the aqueous phase without making contact with the liquid|liquid boundary. Droplets were collected after ten minutes of solution contact and measured for over an hour. Monomodal distributions are observed for all cases with the droplet diameters decreasing with time.

Figure S5: Images for the stability of the observed emulsion over time. A 10 mM HAuCl_4 aqueous solution was pipetted over a 0.1 M $[\text{NBu}_4][\text{ClO}_4]$ DCE solution. The emulsion was allowed to form spontaneously at $t = 0$ min, afterwards overhead stirring was used for 1 min to induce convection and maximize emulsion formation. The observed emulsion was then monitored over time at A) 0 min, B) 2 min, C) 203 min, D) 261 min, and E) 324 min. A noticeable emulsion can be seen up to 261 min showing stability over long periods of time.

Figure S6: Droplet size reproducibility measurements using dynamic light scattering. DCE droplets were spontaneously formed in the aqueous phase when 2 mL of 10 mM H₂AuCl₄ was pipetted above 2 mL of DCE containing 0.1 M [NBu₄][ClO₄]. All droplet dynamic light scattering measurements were taken 5 minutes after contact of both solutions for an N = 4.

	Sample 1	Sample 2	Sample 3	Sample 4	Combined Avg. Size
Average Size (nm)	1230.5	1441.0	1611.8	1283.9	1391.8
Std. Dev. (nm)	135.30	157.11	182.95	187.08	171.72
RSD (%)	10.995	10.902	11.350	14.571	12.337

Table S2: Data and statistical analysis for droplet size and reproducibility measurements using dynamic light scattering (Figure S6). The average size, standard deviation, and relative standard deviation are provided for all samples measured and for a combined average size.

Figure 5: Light microscopy images taken at the water|DCE boundary when the aqueous phase contained 10 mM HAuCl₄ (A), KSCN (B), or K₃[Fe(CN)₆] (C) with either 1 M KCl (i) or 0 M KCl (ii). The DCE phase always contained 0.1 M [NBu₄][ClO₄]. The water and DCE phases are indicated, and the scale bars are 25 μm. (D) Proposed microscopic model of how the transferring ions can cause water-in-oil droplets and/or oil-in-water droplets to form and be stabilized by antagonistic salts. All optical micrographs were taken with a 40x NA 0.60 objective and a 500 ms exposure time.

Figure S11: Light microscopy images taken at the water|DCE boundary when the aqueous phase contained 10 mM HAuCl_4 and the oil phase contained 0.1 M $[\text{NBu}_4][\text{ClO}_4]$ for (A) dichloromethane, (B) nitrobenzene, and (C) chloroform with either 1 M NaCl (i) or 0 M NaCl (ii) present in the aqueous phase. The water and DCE phases are indicated, and the scale bars are 25 μm . All optical micrographs were taken with a 40x NA 0.60 objective and a 500 ms exposure time.

(1) Terry Weatherly, C. K.; Glasscott, M. W.; Dick, J. E. Voltammetric Analysis of Redox Reactions and Ion Transfer in Water Microdroplets. *Langmuir* **2020**, *36* (28), 8231-8239. DOI: 10.1021/acs.langmuir.0c01332.

REVIEWERS' COMMENTS

Reviewer #1 (Remarks to the Author):

The authors have revised their manuscript according to the referee comments. I recommend this manuscript to be published in the revised form.

Reviewer #2 (Remarks to the Author):

I find that the changes made by Colón-Quintana et al. have made the manuscript better. I just have a couple of additional points they could address.

- Now that the manuscript includes SCN⁻ and Fe(CN)₆³⁻ ions, they need to be added to the supporting table S1 as well.
- The mechanistic model proposed in Fig.5D is nice. However, authors have stated that the idea of "water finger" formations at the oil/water interface. If I understood this correctly, this idea comes from simulations in Ref.21. These simulations show water fingers of the size of few angstroms, which involve 2-3 water molecules or so. The protrusions shown in figure 5D (which to a reader would look like "water fingers") are on the order of microns because they are similar size as the droplets. So, the authors need to address/clarify this.

Reviewer #3 (Remarks to the Author):

The authors have undertaken a quite substantial effort to respond to the questions raised in the reviews of the initially submitted work. My earlier impression was that the work was too preliminary for publication together with some fundamental questions on stability and formation mechanism. In the revised paper the authors have now addressed these concerns and have included new information that put the work on solid ground. I believe that the authors' revisions have increased the quality of the paper substantially and that it is now agreeable for publication in Nature Communications.

RESPONSE TO REVIEWERS' COMMENTS

Reviewer #2 (Remarks to the Author):

I find that the changes made by Colón-Quintana et al. have made the manuscript better. I just have a couple of additional points they could address.

- Now that the manuscript includes SCN^- and $\text{Fe}(\text{CN})_6^{3-}$ ions, they need to be added to the supporting table S1 as well.

We thank the reviewer for their thorough review of our manuscript and appreciate this opportunity to clarify. The Gibbs free energies of ion transfer of anions in the aqueous phase are difficult to measure and are overall less important than the partition coefficients, which we report in Supplementary Figures 12 and 13. We have emphasized this in the main text through the following:

“The bulk anionic transfer of chloroaurate from the water to the oil phase necessitates another ion crossing the phase boundary to maintain electroneutrality. Based on ion transfer potentials, the transfer of perchlorate ion from the DCE phase to the aqueous phase would be the most likely process (as opposed to proton or potassium ions transferring from the aqueous phase to the DCE). The Gibbs free energy of ion transfer and the ion transfer potentials for relevant ions in this manuscript (i.e., tetrabutylammonium, tetraethylammonium, hydronium, sodium, potassium, perchlorate, bromide, chloride, and hexafluorophosphate) are provided in Supplementary Table 1. The free energy values for aqueous-phase anions that phase transfer are difficult to measure and are complicated due to the ion pairing mechanism at the boundary; however, their partition coefficients, central to their interfacial flux, can be measured readily and are reported in the Supplementary Information File. This coupled ionic transfer must occur across the liquid|liquid boundary that is formed between these two liquid phases, but few studies have examined what happens at this interface during ion transfer in detail.”

- The mechanistic model proposed in Fig.5D is nice. However, authors have stated that the idea of “water finger” formations at the oil/water interface. If I understood this correctly, this idea comes from simulations in Ref.21. These simulations show water fingers of the size of few angstroms, which involve 2-3 water molecules or so. The protrusions shown in figure 5D (which to a reader would look like “water fingers”) are on the order of microns because they are similar size as the droplets. So, the authors need to address/clarify this.

We thank the reviewer for bringing up this point of confusion. We have added the following sentences to our discussion to be clearer that the initial droplet sizes may be quite small but that the relative size of the droplets compared to the water fingers are not meant to be drawn to scale:

“While the size of the newly formed droplets is difficult to know at the time of their initial formation, these droplets have been observed to coalesce, thereby forming larger

droplets over time. Thus, the initial droplet sizes relative to the water fingers in Figure 6D are not necessarily drawn to scale.”